# Analysis of Issues in Fitness Centers through News Articles before and after the COVID-19 Pandemic in South Korea: Applying Big Data Analysis

**Miyoung Roh** [1] , **Youngchyul Choi** [2] **and Haewon Park** [3,*]

1   College of General Education, Kookmin University, 77 Jeongneung-ro, Seongbuk-gu,
    Seoul 02707, Republic of Korea
2   Faculty of Physical Education, Sangji University, 83 Snagjidae-gil, Wonju-si 26339, Republic of Korea
3   Department of Physical Education, Chungbuk National University, 1 Chungdae-ro, Seowon-gu,
    Cheongju 28644, Republic of Korea
*   Correspondence: gigandbog4@chungbuk.ac.kr

**Abstract:** This study aimed to identify major topics and trends of media reports in news articles related to fitness centers before and after COVID-19 using big data analysis and to suggest future improvements. We collected 20,528 news articles from 2018 to 2019 (pre-COVID-19) and 20,264 news articles from 2020 to 2021 (post-COVID-19) and conducted frequency analysis, topic modeling, and sentiment analysis. The top keywords related to fitness centers were fitness, health, healthcare, and center both before and after COVID-19, but after the outbreak of COVID-19, new keywords emerged, such as digital, COVID-19, indoor, confirmed, platform, and mobile. Since the COVID-19 pandemic, four topics were extracted: COVID-19 and Exercise, Digital Smart Health Care, Health and Beauty, and Indoor Sports Facilities and Quarantine. Sentiment analysis showed that the frequency of negative words increased by approximately three times since the COVID-19 pandemic. Due to COVID-19, the top keywords of the negative data in order were infection, virus, disease, and limit. Based on the identified major issues and trends related to the fitness industry, these results can be used as foundational data for the future planning and policy development of the fitness industry.

**Keywords:** COVID-19; fitness center; topic modeling; sentiment analysis; media big data

## 1. Introduction

Since COVID-19 was first reported in Wuhan, China, in late 2019, it has spread worldwide and brought about major changes in all areas of life, including those related to society, economy, education, industry, and healthcare. Social distancing is an effective countermeasure for preventing the spread of COVID-19, but it has increased the prevalence of non-face-to-face interactions throughout society. As a result, the sports industry was adversely affected to a greater extent than other industries because most sports activities are performed in a face-to-face/contact environment [1,2]. Specifically, indoor sports facilities such as fitness centers or gyms were adversely affected because, unlike in outdoor sports, physical activities were performed in small indoor facilities [2].

In the early period of the COVID-19 pandemic, when the coronavirus was spreading around the world, indoor sports facilities such as fitness centers or gyms were considered high-risk places for coronavirus transmission [3]. In the United States, where the fitness industry is the most developed, the majority of 105,846 health and fitness centers suspended their operations in 2020 due to the COVID-19 pandemic [4]. In Germany, where the fitness industry is most developed among European countries, indoor sports facilities suspended their operations during the COVID-19 period, with the loss incurred by fitness centers due to the decrease in sales expected to be approximately EUR 1.1405 billion [5]. In particular, South Korea's fitness industry has been growing at a rapid pace but to prevent

the spread of coronavirus, from 8 December 2020 to 17 January 2021, the Korean government enforced strict social distancing and indoor gathering restrictions and classified indoor sports facilities as high-risk facilities. As the COVID-19 pandemic prolonged and social distancing became more prevalent, the inflow of new members decreased by half [6]. In contrast, participation in outdoor sports has increased noticeably since the COVID-19 pandemic [7], and consequently, the fitness industry has been facing an unprecedented crisis due to the COVID-19 pandemic.

This crisis has been conveyed to the public most accurately and quickly by news media reports and articles. Furthermore, the accelerated transition to a digital society due to the COVID-19 pandemic made it possible to generate and disseminate a large amount of data in real-time via the Internet. Therefore, press reports and articles are valuable data that can be used to identify the public's overall awareness of this issue and assess the social agenda of the public [8]. Accordingly, the number of studies using big data (e.g., press reports and articles, social media) to analyze health issues related to COVID-19 has increased recently. For example, recent studies investigated issues in the digital healthcare industry through network analysis [9], major issues in news articles related to health communication [10], and COVID-19-related stress factors [11]. However, to our knowledge, no studies have investigated trends or issues considering a general discussion of society using big data analysis based on topics related to fitness centers, which are facing a crisis due to the COVID-19 pandemic.

Existing studies related to COVID-19 and fitness centers have mostly relied on qualitative analysis and survey-based cross-sectional analysis based on limited research topics. Examples of these studies include risk management analysis for sustainable development of fitness centers during COVID-19 [12], exploration of gym-goers' preference for fitness centers during COVID-19 [13,14], and phenomenological analysis of countermeasures for fitness centers [2,15]. Self-reported survey studies have limitations due to the problem of social desirability bias and acquiescence in terms of the response style. Social desirability bias refers to the tendency of respondents to not reveal their honest feelings and behaviors in the survey process, and the acquiescence response style is the tendency of respondents to answer consistently regardless of the content of the question [16]. To analyze the current state of the fitness center industry, which has suffered relatively significant damage due to COVID-19, and promote industry revitalization and future response strategies, fitness center issues before and after the outbreak of COVID-19 need to be clearly identified using big data analysis based on news articles. This study aimed to extract the main keywords and topics from news articles about fitness centers related to COVID-19 and analyze related issues using big data analysis. This study also aimed to identify the variational trends of the main keywords and topics emphasized before and during the COVID-19 pandemic. Moreover, news articles about fitness centers can affect how the public responds, whether positively or negatively, to changes in fitness centers due to COVID-19. Therefore, in this study, we aimed to derive implications for future improvement by detecting positive and negative factors before and during the COVID-19 pandemic through the sentiment analysis of news articles on fitness centers.

The following research questions were addressed:

1. What is the variational trend of the main keywords related to fitness centers before and after the outbreak of COVID-19?
2. What is the variational trend of the major topics related to fitness centers before and after the outbreak of COVID-19?
3. What are the sentiments (positive or negative sentiments) towards fitness centers before and after the outbreak of COVID-19?

The main contributions of this study are as follows: We conducted big data analysis (e.g., topic modeling, sentiment analysis), which is different from previous studies, to explore variational trends in major issues and media reports for fitness centers in South Korea before and during the outbreak of COVID-19. Thus, the results obtained and the big data analysis techniques used in this study may provide guidelines for developing policies

and conducting studies on the fitness industry worldwide. Furthermore, we used online news media articles to quickly discover COVID-19-related issues and the latest government policies and regulations for fitness centers, enabling fitness center owners and business executives to contribute to diagnosing and improving field issues in the post-pandemic era. Furthermore, we used online news media articles to quickly discover COVID-19-related issues and the latest government fitness center policies and regulations that may support fitness center owners and business executives in resolving field issues in the post-pandemic era. Therefore, the findings can contribute to establishing sustainable policies to ensure proper operation of fitness centers and provide a context for discussing the post-pandemic era for those who work in fitness centers.

## 2. Literature Review

### 2.1. Studies on Fitness Centers during COVID-19

The COVID-19 pandemic has brought about significant changes to daily lifestyles but also to the sports industry environment. Many studies have been conducted on the risk management and response challenges of the sports industry due to COVID-19 [2,12,17]. In particular, compared with outdoor sports, indoor fitness centers or gyms have been affected considerably more by the pandemic [2]. To overcome this, several studies have investigated the COVID-19 impact on indoor fitness centers [18–20]. Myers et al. [2] observed and interviewed gym owners, trainers, and clients, focusing on the impact of COVID-19 on those involved in fitness centers. The authors noted that major concerns emerged regarding the substantial operational and financial difficulties for owners given the easy transmission of coronavirus among individuals in overpopulated indoor spaces. However, individuals still tried to maintain physically active participation via online training programs, which have emerged as a new fitness trend during the pandemic. Similarly, according to a worldwide survey conducted to determine fitness trends for 2021 with the participation of more than 4,500 health and fitness professionals [21], the COVID-19 pandemic has changed the 2021 fitness trends, and the new no. 1 trend was online training (no. 26 in 2020), with outdoor activities (no. 4) and virtual training (no. 6) becoming increasingly popular. By contrast, group training (no. 17), which was no. 3 in 2020, fell dramatically, probably due to gyms closing or the recommendations to limit group gatherings owing to social distancing policy. Park and Kwon [12] analyzed sustainable crisis management planning for fitness centers using importance–performance analysis. They found that social distancing between employees and members was a factor that needs to be maintained and strengthened for sustainable crisis management of fitness centers during the COVID-19 pandemic. However, regular disinfection of fitness centers indicated high importance but low performance, confirming that a systematic regular disinfection plan including cleaning gym equipment before and after use should be established to prevent transmission of coronavirus. These results were consistent with a study conducted before a lockdown announcement in Saudi Arabia in February 2020 where a lack of gym members' awareness of safety precautions was identified as a problem to be addressed [18]. Accordingly, after the COVID-19 outbreak, fitness center executives and employees were facing various challenges and changes, and efforts have been made to determine countermeasures through surveys and qualitative research.

Most previous studies on behavioral changes of gym attendance due to COVID-19 and on risk management in fitness centers have relied on surveys or in-depth interviews with gym owners, trainers, and members. Looking at some related studies, Park and Kwon [12] conducted a survey of fitness center executives and managers in South Korea to verify the importance and performance of sustainable risk management for fitness centers. Ong et al. [13] analyzed the preference of gym-goers in fitness centers in the Philippines during COVID-19, and in another study, Ong et al. [14] investigated the factors influencing gym-goers' behavioral intentions towards fitness centers based on a self-report questionnaire. Some further examples of the qualitative research include the following: Myers et al. [2] investigated the effects of COVID-19 on fitness centers from two perspec-

tives, that of gym owners and clients, through interviews and observations. Kwon and Nam [15] explored the crisis response and coping process of fitness centers during COVID-19 through in-depth interviews with fitness center owners and business executives. To explore the physical and psychological change of fitness center members due to COVID-19, Kaur et al. [22] conducted semi-structured telephone interviews with adults who regularly exercised before the pandemic to explore how they coped with staying physically active during the lockdown period when gym and fitness centers were closed.

The self-report survey studies conducted thus far suffer from social desirability bias, which means that respondents may not reveal their honest feelings and behavioral status during the survey process, and from the acquiescence response style [16], which means that they tend to respond consistently regardless of the question content. Moreover, there is a limit to identifying the practical issues of fitness centers due to COVID-19 using only research based on surveys or interviews conducted with convenient participant samples. By contrast, investigations through mass media reports have the advantage of grasping the general perception of the public on issues and social agenda [8], and despite the recent increase in studies using big data to analyze health issues related to COVID-19 [9–11], research in the fitness center industry remains very limited.

### 2.2. Impact on Korean Fitness Centers during COVID-19

An increasing number of fitness centers and gyms have opened as living a healthy lifestyle has become popular worldwide [23]. The fitness industry in South Korea was also growing rapidly until the outbreak of COVID-19. Despite the COVID-19 pandemic, approximately 10,000 fitness centers were operating nationwide in 2020, corresponding to an increase of approximately 54% over the past ten years [24]. According to the 2016 International Health Racquet and Sports Club Association (IHRSA) report [23], the number of fitness centers in South Korea ranked seventh globally (6839) and first among Asian countries. However, since the outbreak of COVID-19, the South Korean government implemented and enforced strict social distancing for 41 days between 8 December 2020 and 17 January 2021. Consequently, indoor sports facilities, classified as high-risk, were forcibly closed [25]. The results of a survey on the damage in the sports industry caused by COVID-19 conducted in April 2020 targeting 3000 sports companies nationwide indicated that in most industries sales decreased significantly compared to the previous year, with the sports service industry decreasing the most by 84.14%, followed by the sports facility industry (61.4%) and the sports goods industry (51%) [26]. In particular, due to social distancing, the sales of indoor sports facilities, such as fitness and taekwondo centers, showed a decrease of 91.3% and 81.0%, respectively. One year after the COVID-19 outbreak, in an online survey of 988 indoor sports facility owners conducted by the COVID-19 Indoor Sports Facility Emergency Response Committee, 99% of indoor sports facility businesses showed a decrease in sales due to COVID-19, and 59.7% showed that their rent was overdue, indicating that they were facing an intensive crisis [27].

Moreover, a study conducted by Bae et al. [19] investigated the epidemiological characteristics of the COVID-19 outbreak at fitness centers in Cheonan, Korea, from 24 February to 13 March 2020. According to the results, 116 cases (8 instructors, 57 Zumba class enrollees, 37 family members, and 14 others) related to Zumba classes in fitness centers were confirmed as of March 11. The study reported that Zumba classes may provide an environment of easy transmission of coronavirus because droplets produced by the participants' exhalation or coughing during high-intensity aerobic exercise in a crowded indoor space may reach the nose, mouth, and eyes of the other participants. As this information was extensively reported in the mass media, fitness centers and indoor sports facilities were recognized by the public as high-risk facilities for the spread of the coronavirus, causing further disruption to their operations. Kwon and Nam [15], in another study, interviewed fitness center representatives and managers to explore their experiences and coping process during the COVID-19 outbreak. The interviewees expressed that new clients were reluctant to visit fitness centers, as news media reported that fitness centers were dangerous due



to droplets and fomites generated during exercise. Thus, mass media, such as real-time online news articles with a large amount of data, can have a significant impact on the behavior of the public. In this regard, it is necessary to understand the media coverage of the fitness centers focusing on changes since COVID-19 through big data analysis of news media articles.

### 2.3. Impact of News Media on the Public during COVID-19

News media reflects various issues such as economy, society, and culture that the public encounters and understands as major issues [28]. News media focuses on the function of reporting objective facts and conveying information, and its main role is to produce and disseminate what the public needs to know, which influences public opinion and structures people's epistemology [29]. As such, news media can have a significant impact on the public. Specifically, they modify people's behaviors and attitudes based on the content of the news articles [30]. According to Anwar et al. [30], they analyzed the impact of mass media and public health communications on the public during the early onset of COVID-19. They indicated that the role of the mass media in influencing the public's behavior became more crucial by providing the public with the most accurate and prompt information about the novel coronavirus infection and its prevention guidelines (e.g., hand washing, use of masks, and social distancing) that can have a positive impact on preventing the spread of disease through live updates news media articles. In contrast, a study conducted by Kwon and Nam [15] found a negative impact of news media to the public. The results indicated that news media frequently reported fitness centers as high-risk facilities for the spread of the coronavirus, causing further disruption to operations of fitness centers. Thus, mass media such as real-time online news articles with vast amounts of data can significantly influence public behavior and society.

COVID-19 accelerated the transition to the age of digital society, and social media engagement has increased, generating a large amount of data in real time [31]. Furthermore, currently, the Internet is a considerable information source for the public. In particular, online mass media news is a key source for quickly finding issues and updated governmental policies and regulations related to COVID-19 [32,33]. Online news articles, which are unstructured big data, have the advantage of being digitalized and analyzable [34]. Moreover, news big data is useful to comprehensively analyze and understand the reality, such as understanding the contents related to the news from various perspectives and the current issues of stakeholders and solutions [35]. Consequently, studies using online news media articles have been actively conducted in various fields such as education [8], leisure activity [36], and health communication [10] during COVID-19 to predict various trends and diagnose the direction of change by reflecting the issues and events of the times contained in news materials.

### 2.4. Big Data Analysis in the Health and Fitness Industry

The term big data refers to analytics that can generate trends and patterns through various analysis methods for understanding specific industry [31] with a large amount of data beyond the range that can be collected, stored, and analyzed within traditional database software [37]. The concept goes beyond large-volume data and helps to understand social phenomena and predict the future through the correlation of the data [38]; as such, it can be suitable for understanding the general perception and trend changes of the public rather than a limited group.

After the COVID-19 outbreak, people have increased their interest in health [13], but many researchers are concerned that the closing of fitness centers may adversely affect individuals' health [22,39,40], which may trigger market and trend changes in the fitness industry. Consequently, in the health and fitness industry, studies have been recently conducted to explore key topics and trend changes during COVID-19 using big data analysis such as text mining and topic modeling. Text mining techniques are widely used to analyze news media trends [41]. Text mining is an analysis technique that converts a large amount

of unstructured text data so that they can be analyzed to derive meaningful patterns or topics latent in the text [42]. Topic modeling is an algorithm-based text mining technique that extracts key topics hidden in a large amount of unstructured text data [43]. Kim [36] conducted topic modeling on media big data before and after COVID-19 and investigated changes in the trends of leisure industry in South Korea. The topic modeling results indicated that before COVID-19, leisure tourism, safety, business, programs, and sports were extracted as topics, whereas with the COVID-19 outbreak, new topics such as increased leisure time and online leisure business emerged. The author noted that the media extensively reported on outdoor activities, such as hiking and camping, which are considered safe against coronavirus transmission, as an alternative to staying at home. In a similar vein, several studies have conducted big data analysis on media news articles to explore the rapidly growing trend of home training in South Korea due to COVID-19 [44–46]. Many researchers have raised concerns about the increase in physical inactivity and sedentary lifestyle due to strict COVID-19 quarantines [47–49]. To investigate this, Ding et al. [39] used Google Trends data to explore community interest in physical activity before and after the COVID-19 outbreak in the USA, the UK, and Australia from June 2019 to May 2020. The results indicated that people's interest in physical activity in all countries surged and peaked in April 2020 during lockdown, then fell but stayed higher than before the COVID-19 pandemic. This finding may not be the result of behavioral change regarding physical activity, but it indicates that COVID-19 led individuals to have more exercise intention. The authors explained the potential reason for the increase in exercise interest may be driven by the recommendation to be physically active from media, governments, and health authorities. Furthermore, attempts were made during the COVID-19 pandemic to estimate the pandemic's future trend using customized models [50,51] and to explore accurate diagnosis of COVID-19 using deep learning [52].

Sentiment analysis is a widely used technique to analyze people's opinions by recognizing the relationship between the words in the text and to quantify their sentiment as positive or negative [53]. For example, Şahin, Gümüş, and Gençoğlu [54] analyzed Twitter data related to physical activity with "physical" or "fitness" or "gymnastics" as a keyword during the early stage of COVID-19 using Google Cloud Natural Language, which can identify people's sentiment from text. They found an increase in the number of both positive and negative tweets during COVID-19 compared with the non-COVID-19 period. The results indicated that people were actively using online social media to be physically active as fitness centers and gyms were closed during the lockdown. Moreover, a big data analysis was conducted to examine research trends in published journal articles related to the fitness industry. Sung [55] applied text mining and topic modeling to obtain the research trends of the Korean fitness industry from 2011 to 2021 and compared the first half (2011–2014) and the second half (2015–2021). The results suggested that customers, companies, smart care, wearable devices, and applications have appeared as new keywords in articles since 2015. Most research topics concerned training in the first half while commerciality and smart care emerged in the second half, showing the expansion of the scope of fitness-center-related research.

Studies applying big data analysis have been conducted on health-related issues that have undergone changes due to COVID-19. The research topics have focused on exploring trends and issues at one time point, and the results were primarily analyzed by using topic modeling. However, big data analysis research on the sustainability of fitness centers, which have been considerably impacted by the COVID-19 outbreak, is limited. Therefore, research to expand a longitudinal study divided into two time points, before and after COVID-19, is required, with a sentiment analysis based on news articles to verify positive and negative sentiment aspects for deriving implications for future improvement. In this study, we identify the trend of changes in keywords, topics, and sentiment aspects of fitness centers before and after the pandemic using topic modeling and sentiment analysis to provide implications for sustainability in the field.

## 3. Materials and Methods

### 3.1. Data Collection

We collected news articles based on BIGKinds (Korea Integrated News Database System), a news big database system, provided by the Korea Press Foundation. Since BIGKinds automatically collects and classifies news articles from 54 media outlets in real-time and provides them to the public, it is widely used in the big data analysis field using news articles [56]. In this study, news articles from 54 news daily outlets provided by BIGKinds, including general daily newspapers, economic journals, regional daily newspapers, and broadcasting companies, were set as data collection targets. Articles were extracted on a monthly basis with a combination of three search keywords, "Fitness" and ("Center" or "Club"). To compare two points in time, we defined the period from 1 January 2018 to 31 December 2019 as pre-COVID-19 and 1 January 2020 to 31 December 2021 as post-COVID-19. After data collection, advertisements, publicity articles, and news articles that were not related to fitness centers were excluded from big data analysis through consultation among researchers. A total of 40,792 news articles were searched: 20,528 pre-COVID-19 news articles and 20,264 post-COVID-19 news articles using the BeautifulSoup library of Python version 3.10.4.

### 3.2. Data Preprocessing

The collected research data need to be processed into a form that can be analyzed to ensure proper data utilization; for this purpose, preprocessing must be performed. Preprocessing involves converting the language used by humans into data that computers can understand. In this process, unnecessary data is separated and removed from the collected research data, with the remaining research data processed to suit the analysis method and purpose. To analyze unstructured text data, data preprocessing was performed to subdivide morphemes and remove stop words [57]. For the review text data, morphological analysis was performed using Mecab of the Python Konlpy package, widely used in natural language processing (NLP). Then, we performed tokenization to extract key nouns, adjectives, and verbs. After that, unnecessary phrases and words, numbers with low relevance, special symbols, punctuation marks, and unidentified words were deleted. In addition, researchers repeatedly performed the task of correcting words with postpositions.

### 3.3. Data Analysis

#### 3.3.1. Term Frequency

We conducted a term frequency (TF) analysis to determine the frequency of occurrence of specific words in an entire document. TF analysis is a technique that prioritizes words extracted after the text preprocessing process and determines the importance of a specific word based on the number of occurrences. TF analysis is the most common and basic analysis technique used in text mining analysis. In particular, TF analysis is also the most basic technique first used in text mining to understand data flow. An analysis based on TF indicates the frequency of occurrence for specific words in an entire document. The TF indicates the number of times a specific word appears in a document, with a higher TF value indicating that a word is used more often in the document [58]. Therefore, a word with a high frequency of occurrence is generally a word highly related to the research topic while also likely containing connotations and being a keyword of the research topic.

#### 3.3.2. Topic Modeling

We derived appropriate topics for classifying fitness-center-related data using the topic modeling technique based on Latent Dirichlet Allocation (LDA). Topic modeling is a method of extracting meaningful topics from a large amount of unstructured text data and groups words with high relevance to infer a topic that can represent the group [43]. Among the various algorithms of topic modeling (e.g., Latent Semantic Indexing (LSI), Probabilistic Latent Semantic Analysis (pLSA), Latent Dirichlet Allocation (LDA), LDA is the most commonly used method in research in various fields [59]. LDA is a method of reducing

probabilistically highly related words into n topics in a large amount of unstructured text data, and since it is calculated based on the probability rather than the frequency of occurrence of a specific word in texts, the meaning of the topic is further enhanced [60]. In this study, we conducted topic modeling using the gensim and sklearn libraries of Python. We used the pyLDAvis module to select the most appropriate number of topics, considering the range in which topics do not overlap and the range that can optimize the coherence and perplexity values, which indicate the degree of association between words within the topic. In this study, the statistical evaluation was considered along with the validity and possibility of interpretation of the results of the topics derived in the process of selecting the optimal number of topics.

We also considered the relevance of the derived topics to the research problem in this selection process. In fact, it is not desirable to select the optimal number of topics by simply using a statistical method. Instead, it is better to select the optimal number of topics by determining how meaningful and analytically useful the derived topics are according to the research problem. Therefore, we believe it is rational to consider the abovementioned factors when selecting the optimal number of topics [61].

### 3.3.3. Sentimental Analysis

We conducted a sentiment analysis on fitness-center-related news articles utilizing a sentiment dictionary. We used preprocessed refined data in the sentiment analysis and calculated the ratio of positive and negative keywords using the Textom sentiment dictionary based on Korean text. Then, the frequency value of words was derived for each document using the CountVectorizer module of Python's scikit-learn library, with positive and negative sentiment words extracted for each individual collection unit.

## 4. Results

### 4.1. Keyword Analysis from Term Frequency

Table 1 shows the results of the keyword analysis from term frequency and represented the 25 most frequent words and their frequency values in pre-COVID-19 and post-COVID-19, respectively. The top four most frequent words in common were fitness (pre-COVID-19, n = 55,426; post-COVID-19, n = 54,285), health (pre-COVID-19, n = 52,888; post-COVID-19, n = 52,465), healthcare (pre-COVID-19, n = 35,441; post-COVID-19, n = 32,738), and center (pre-COVID-19, n = 9124; post-COVID-19, n = 10,247) both before and after the outbreak of COVID-19, and the following words also appeared to be included in the top 25 frequent words: weight training (pre-COVID-19, n = 5645; post-COVID-19, n = 6359), South Korea (pre-COVID-19, n = 6544; post-COVID-19, n = 4074), industry (pre-COVID-19, n = 5421; post-COVID-19, n = 4469), exercise (pre-COVID-19, n = 4299; post-COVID-19, n = 4450), Seoul (pre-COVID-19, n = 4485; post-COVID-19, n = 4045), medical (pre-COVID-19, n = 3552; post-COVID-19, n = 4286), competition (pre-COVID-19, n = 4166; post-COVID-19, n = 3518), release (pre-COVID-19, n = 2509; post-COVID-19, n = 2948), professional (pre-COVID-19, n = 9124; post-COVID-19, n = 10,247), and management (pre-COVID-19, n = 2280; post-COVID-19, n = 2818). After the outbreak of COVID-19, the new frequent words that appeared in the high frequency were digital (n = 7762), COVID-19 (n = 6575), service (n = 6165), indoor (n = 4582), smart (n = 3276), development (n = 3206), confirmed (n = 3200), platform (n = 2733), use (n = 2649), and mobile (n = 2468). These results indicated that news media and public interest related to fitness centers have changed in post-COVID-19 compared to pre-COVID-19.

**Table 1.** Keyword analysis from Term Frequency (Rank Order).

| Rank | Pre-COVID-19 (2018–2019) | | Post-COVID-19 (2020–2021) | |
|---|---|---|---|---|
| | Word | Frequency | Word | Frequency |
| 1 | fitness | 55,426 | fitness | 54,285 |
| 2 | health | 52,888 | health | 52,465 |
| 3 | healthcare | 35,441 | healthcare | 32,738 |
| 4 | center | 9124 | center | 10,247 |
| 5 | South Korea | 6544 | digital | 7262 |
| 6 | weight training | 5645 | COVID-19 | 6575 |
| 7 | industry | 5421 | weight training | 6359 |
| 8 | Seoul | 4485 | service | 6165 |
| 9 | exercise | 4299 | indoor | 4582 |
| 10 | competition | 4166 | industry | 4469 |
| 11 | medical | 3552 | exercise | 4450 |
| 12 | golf | 2850 | medical | 4286 |
| 13 | sport | 2672 | South Korea | 4074 |
| 14 | global | 2603 | Seoul | 4045 |
| 15 | release | 2509 | golf | 4037 |
| 16 | club | 2501 | competition | 3518 |
| 17 | bodybuilder | 2405 | smart | 3276 |
| 18 | professional | 2385 | development | 3206 |
| 19 | management | 2280 | confirmed | 3200 |
| 20 | brand | 2153 | release | 2948 |
| 21 | hold | 2083 | professional | 2887 |
| 22 | product | 2059 | management | 2818 |
| 23 | information | 2043 | platform | 2733 |
| 24 | beauty | 2008 | use | 2649 |
| 25 | swimming pool | 1946 | mobile | 2468 |

*4.2. Topic Modeling Analysis*

A topic modeling analysis was conducted on fitness-center-related news articles at two time points before and after the outbreak of COVID-19. As a result, six and four main topics were extracted in pre-COVID-19 and post-COVID-19, respectively. We identified the main topics in two time periods and extracted the top 10 weighed words for each topic.

Table 2 shows the topic modeling results for fitness-related news articles and their frequency ratios before the outbreak of COVID-19. The main topics before COVID-19 can be described as Health and Beauty (Topic 1), Digital Healthcare (Topic 2), Bio-Healthcare Industry (Topic 3), Community Health Promotion (Topic 4), Business and Sales (Topic 5), and domestic Bio-Health Industrial Complex (Topic 6). As shown in Table 2, Topic 1 consists of keywords such as exercise, weight training, beauty, profit, investment, media, electronics, and trainer, which were related to health and beauty. Topic 2 consists of keywords of healthcare, medical, digital, service, development, health, smart, hospital, mobile, and product, which were related to the digital healthcare industry. Topic 3 consists of keywords such as bio, healthcare, industry, new drug, government, development, pharmaceutical, innovation, and research, which were related to the biopharmaceutical industry. Topic 4 consists of keywords such as health, convention, public health center, disease, Seoul, university, district, support, and center, which were related to health promotion projects in local communities. Topic 5 consists of keywords such as business, release, robot, medicine, children, Seoul, pharmacy, budget, and sales, as these words are related to business and sales. Finally, Topic 6 consists of keywords such as center, Daegu, city, Chungju, government expenses, senior, industrial complex, and district, which were related to the domestic bio-health industrial complex.

**Table 2.** Topic modeling result before COVID-19: Top 10 word probabilities to 6 topics.

| Category | Topic 1 | Topic 2 | Topic 3 | Topic 4 | Topic 5 | Topic 6 |
|---|---|---|---|---|---|---|
| Rank | **Health and Beauty** | **Digital Healthcare** | **Bio-Healthcare Industry** | **Community Health Promotion** | **Business and Sales** | **Bio-Health Industrial Complex** |
| 1 | exercise | healthcare | bio | health | business | center |
| 2 | weight training | medical | healthcare | convention | release | Daegu |
| 3 | beauty | digital | industry | public health center | robot | city |
| 4 | profit | service | new drug | disease | medicine | Chungju |
| 5 | investment | development | government | Seoul | boss | government expenses |
| 6 | media | health | development | university | children | golf |
| 7 | electronic | smart | pharmaceutical | district | Seoul | senior |
| 8 | trainer | hospital | innovation | support | pharmacy | industrial complex |
| 9 | committee | mobile | forum | center | budget | candidate |
| 10 | professor | product | research | hold | sales | district |
| Proportions (%) | 21.9 | 21.5 | 20.6 | 20.4 | 8.9 | 6.7 |

The highest proportions of these topics before COVID-19 in descending order are Health and Beauty (24.9%), Digital Healthcare (21.5%), Promotion of Health in Local Communities (20.4%), and Bio-Healthcare followed by Business and Sales and Bio-Health Industrial Complex (6.7%). These topics are focused on the health and fitness industry. The fitness industry in South Korea was growing rapidly until the outbreak of COVID-19. Fitness is not simply a concept limited to weight training or bodybuilding but refers to an industry that includes not only direct physical training through exercise but also healthcare, clothing, meals, and exercise goods [55]. Therefore, until the outbreak of COVID-19, most of the fitness-center-related news articles were about topics related to the pursuit of promoting public health and wellness, which is the essential value provided by the fitness industry. Moreover, Digital Healthcare (Topic 2) and Bio-Healthcare Industry (Topic 3) can be seen as topics that the fourth industrial revolution and digital society transformation are rapidly industrializing in the health and fitness industry.

Table 3 shows the topic modeling results for fitness-related news articles and their frequency ratios after the outbreak of COVID-19. The main topics during the COVID-19 pandemic can be interpreted as COVID-19 and Exercise (Topic 1, 38.5%), Digital Smart Health Care Service (Topic 2, 24.8%), Health and Beauty (Topic 3, 21.9%), and Indoor Sports Facilities and Quarantine (Topic 4, 14.8%). As shown in Table 3, Topic 1 consists of keywords such as COVID-19, confirmed, infection, vaccine, export, exercise, approval, case, and group, which were related to the situation of the fitness industry after the COVID-19 outbreak. Topic 2 consists of keywords such as healthcare, digital, medical, service, development, health, research, smart, and platform, which were related to IT-based digital smart healthcare services that have drawn attention since the COVID-19 pandemic. Topic 3 consists of keywords of bio, weight training, fitness, life, information, beauty, brand, global, release, and healthcare, which were related to maintaining and promoting health and beauty. Finally, Topic 4 consists of keywords such as center, indoor, physical training, fitness, use, pass, quarantine, and indoor facility, which were related to the use of indoor sports facilities and preventive measures to prevent the spread of COVID-19.

**Table 3.** Topic modeling result after COVID-19: Top 10 word probabilities to 4 topics.

| Category | Topic 1 | Topic 2 | Topic 3 | Topic 4 |
|---|---|---|---|---|
| **Rank** | **COVID-19 and Exercise** | **Digital Smart Healthcare Service** | **Health and Beauty** | **Indoor Sports Facilities and Quarantine** |
| 1 | COVID-19 | healthcare | bio | center |
| 2 | confirmed | digital | weight training | indoor |
| 3 | infection | medical | fitness | physical training |
| 4 | vaccine | service | life | fitness |
| 5 | export | development | information | golf |
| 6 | exercise | healthcare | beauty | use |
| 7 | approval | research | brand | pass |
| 8 | case | data | global | quarantine |
| 9 | group | smart | release | swimming pool |
| 10 | district | platform | healthcare | indoor facility |
| Proportions (%) | 38.5 | 24.8 | 21.9 | 14.8 |

Before and after COVID-19, Health and Beauty and Health Care Service appeared in both cases. This result indicated that the media was concerned with healthcare services, a health and fitness-related industry, to promote the health and welfare of the public, which is the key role of fitness centers regardless of the COVID-19 pandemic.

Specifically, new topics, including COVID-19 and Exercise (Topic 1) and Use of Indoor Sports Facilities and Preventive Measures to Limit the Spread of COVID-19 (Topic 4), became the biggest issues and represented 53% of all topics. Therefore, there has been a change in the fitness industry due to COVID-19. The main keywords for Topic 1 were COVID-19, confirmed, infection, vaccine, exercise, approval, and group. Topic 2 included main keywords such as center, indoor, physical training, fitness, use, pass, and quarantine. Therefore, with gatherings being banned in fitness centers after the outbreak of COVID-19, there were concerns about infection occurring when exercising in indoor spaces where many people gather. In addition, excessively strict preventive measures (e.g., COVID passports, checking whether people have tested negative via PCR tests) imposed on indoor sports facilities drew the public's attention. These findings revealed different demand and changes in the fitness industry due to COVID-19.

*4.3. Sentiment Analysis*

We conducted a sentiment analysis on fitness-center-related news articles before and after the outbreak of COVID-19, and the results are shown in Table 4. Figure 1 shows the visualization of the result of sentiment analysis before and after the outbreak of COVID-19: top 15 negative and positive words.

Considering fitness-center-related news articles before the outbreak of COVID-19, the frequency of positive words was high, while the frequency of negative words was low (Figure 1a). Conversely, the frequency of positive words in fitness-center-related news articles after the outbreak of COVID-19 decreased slightly compared to before COVID-19, but the frequency of negative words increased by about three times (Figure 1b). In fact, the ratio of the top 15 positive and negative words before COVID-19 was 72.11% for positive words and 27.89% for negative words. In contrast, the ratio of the top 15 positive and negative words after COVID-19 was 54.37% for positive words and 45.63% for negative words. These results can be attributed to the increase in the appearance of negative words after COVID-19.

**Table 4.** Sentiment analysis result.

| | Pre-COVID-19 (2018–2019) | | | | | Post-COVID-19 (2020–2021) | | | |
|---|---|---|---|---|---|---|---|---|---|
| Rank | Positive Word | Frequency | Negative Word | Frequency | Rank | Positive Word | Frequency | Negative Word | Frequency |
| 1 | rise | 1815 | violence | 524 | 1 | reinforcement | 1049 | infection | 1769 |
| 2 | possible | 1501 | danger | 366 | 2 | rise | 881 | virus | 853 |
| 3 | profit | 836 | obstacle | 325 | 3 | increase | 731 | disease | 682 |
| 4 | interest | 654 | decrease | 298 | 4 | profit | 718 | infectious disease | 644 |
| 5 | popularity | 609 | charge | 285 | 5 | treatment | 718 | rejection | 558 |
| 6 | progress | 608 | argument | 271 | 6 | activation | 645 | ban | 509 |
| 7 | success | 555 | suspicion | 257 | 7 | interest | 582 | declaration | 439 |
| 8 | expectation | 519 | damage | 252 | 8 | safety | 542 | limit | 425 |
| 9 | improvement | 516 | problem | 246 | 9 | luxury | 526 | danger | 415 |
| 10 | famous | 449 | chronic | 215 | 10 | luck | 523 | discontinuation | 379 |
| 11 | activation | 449 | downturn | 205 | 11 | continuation | 512 | damage | 276 |
| 12 | benefit | 435 | pain | 199 | 12 | challenge | 505 | chronic | 245 |
| 13 | creation | 428 | injury | 162 | 13 | progress | 505 | exclusion | 241 |
| 14 | excellent | 347 | burden | 157 | 14 | remedy | 498 | obstacle | 239 |
| 15 | enjoyment | 344 | worry | 130 | 15 | popularity | 474 | problem | 223 |

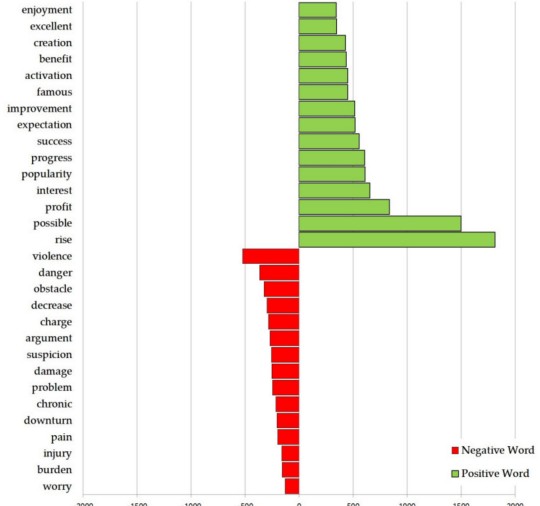

(**a**) Negative and positive word before COVID-19

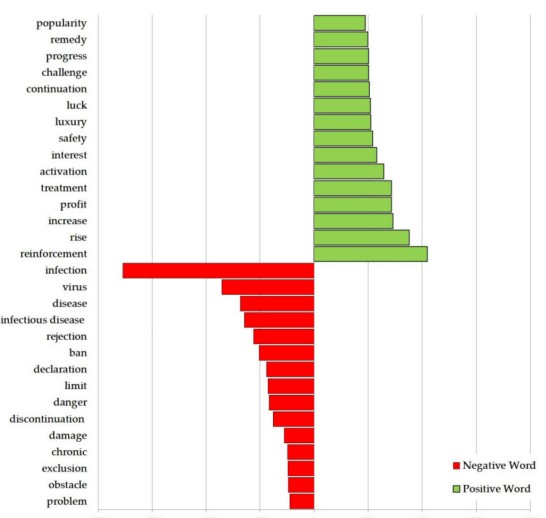

(**b**) Negative and positive word after COVID-19

**Figure 1.** Sentiment analysis result of visualization before (**a**) and after (**b**) COVID-19. *X*-axis: top 15 negative (red bar) and positive words (green bar). *Y*-axis: The frequency of words.

We examined the positive words: specifically, rise (pre-COVID-19, n = 1815; post-COVID-19, n = 881), profit (pre-COVID-19, n = 836; post-COVID-19, n = 718), interest (pre-COVID-19, n = 654; post-COVID-19, n = 582), popularity (pre-COVID-19, n = 609; post-COVID-19, n = 474), progress (pre-COVID-19, n = 608; post-COVID-19, n = 505), and activation (pre-COVID-19, n = 449; post-COVID-19, n = 645) appeared with high frequency both before and after the outbreak of COVID-19. In contrast, reinforcement (n = 1049), increase (n = 731), treatment (n = 718), safety (n = 542), luck (n = 523), continuation (n = 512), challenge (n = 505), and remedy (n = 498) were new positive words that appear with high frequency after the COVID-19 outbreak. We examined the negative words: specifically, danger (pre-COVID-19, n = 366; post-COVID-19, n = 415), obstacle (pre-COVID-19, n = 325; post-COVID-19, n = 239), damage (pre-COVID-19, n = 252; post-COVID-19, n = 276), problem (pre-COVID-19, n = 246; post-COVID-19, n = 223), and chronic (pre-COVID-19, n = 215; post-COVID-19, n = 245) appear with high frequency before and after the outbreak of COVID-19. In contrast, infection (n = 1769), virus (n = 853), disease (n = 682), infectious disease (n = 644), rejection (n = 558), ban (n = 509), declaration (n = 439), limit

(n = 425), discontinuation (n = 379), and exclusion (n = 241) were new negative words that appeared with high frequency after the COVID-19 outbreak. These results seem to have been discussed by the public opinion that restrictions on the operation of fitness centers and suspension of business would directly lead to restrictions on participation in exercise, which may negatively affect public health as well as society as a whole.

## 5. Discussion and Implications

With the COVID-19 pandemic arousing public concern worldwide, non-face-to-face interactions have particularly increased in frequency and have continued throughout society since the outbreak of COVID-19. As a result, the fitness center industry was more adversely affected than other industries considering most sports activities are performed in a face-to-face environment. Therefore, in this study, we focused on fitness centers in relation to COVID-19 and utilize topic modeling to extract main keywords and topics based on fitness-center-related news articles before and after COVID-19. Then, we analyzed the keywords and topics to identify the key issues contained in the press reports and articles. Furthermore, we identified the trend of changes in society in general based on the changes in the main keywords and topics emphasized before and after the outbreak of COVID-19. Moreover, we derived implications for the future improvement of the fitness center industry by detecting positive and negative factors before and after COVID-19 through the sentiment analysis of news articles related to fitness centers.

First, the results of keyword analysis showed that the main keywords that appeared in major domestic press reports, such as fitness, health, healthcare, center, and exercise, frequently appeared before and after COVID-19. Moreover, regardless of COVID-19, many of the fitness-center-related news articles were reported as an exercise place to maintain individuals' health. Notably, key words such as digital, COVID-19, indoor, smart, development, confirmed, platform, use, and mobile have newly emerged since the outbreak of COVID-19, possibly indicating a movement to overcome members' limited access to indoor facilities caused by increases in the number of confirmed cases of COVID-19. Furthermore, this result can also be interpreted positively to indicate an increased interest of the press and public regarding the development of new services, such as platforms that utilize digital and mobile technologies. The findings may indicate an active effort to continually identify methods to diversify consumer participation in exercises. Therefore, the press and public have shown interest in finding alternatives to continually participate in physical exercise despite restrictions on using indoor sports facilities due to COVID-19, thus promoting changes in the trend of the fitness industry.

In contrast, keywords for owners and trainers who work at fitness centers appeared relatively less frequently. Therefore, more attention and support are needed for the owners and trainers in these facilities. In the early period of the COVID-19 pandemic, fitness centers for indoor sports were forcibly closed because they were classified as high-risk facilities for coronavirus infection. As a result, new members joining fitness facilities declined by half in South Korea, resulting in fitness facilities experiencing financial difficulties [6]. Consequently, many fitness centers suspended their operations or closed entirely, resulting in the mass unemployment of sports instructors [62]. Moreover, online home training systems for sports-related activities have been upwardly trending [2,6], placing an additional burden on fitness centers. The Ministry of Culture, Sports, and Tourism selects companies affected by COVID-19 and provides support measures to expand sports company loans. Furthermore, the Ministry of Culture, Sports, and Tourism provided emergency relief funds for workers in the sports industry. However, the sales in most industries have decreased significantly compared to the previous year. In particular, sales for businesses such as fitness centers that have been recommended to suspend their operations according to social distancing measures have plummeted by 91.3% compared to the previous year [26]. Considering these findings, it is questionable whether these policies are effective for those working on the frontline. Therefore, it is necessary to understand the opinions and demands of those who work in fitness centers to identify effective support measures from a mid-to-long-term

perspective to ensure these support measures may benefit both the workers and consumers in the fitness industry.

Second, the topics that appeared in media reports were analyzed using topic modeling. Based on the analysis results, four topics were determined related to fitness centers after COVID-19: COVID-19 and Exercise (Topic 1), Digital Smart Health Care Service (Topic 2), Health and Beauty (Topic 3), and Use of Indoor Sports Facilities and Preventive Measures to Limit the Spread of COVID-19 (Topic 4). These topics are focused on exercise and health-related industries. We examined the similarities and differences between fitness-center-related topics before and after COVID-19; Health and Beauty and Health Care Service appeared in both cases. Moreover, the media was concerned with healthcare services, a health-related industry, to promote the health and welfare of the public, which is the key role of fitness centers regardless of the COVID-19 pandemic.

Before the outbreak of COVID-19, most of the media reports related to fitness were about topics related to the pursuit of promoting public health, which is the essential value provided by the fitness industry. However, after the outbreak of COVID-19, new topics, including COVID-19 and Exercise and Indoor Sports Facilities and Quarantine, became the biggest issues. When the government imposed strict social distancing measures in the early period of the COVID-19 pandemic, facilities like restaurants, cafes, and grocery stores operated their businesses in a limited manner. However, indoor sports facilities were either entirely forcibly closed or had to operate their business with strict preventive measures (e.g., shortened operations, increased distance between exercise machines, limited occupancy, limiting the speed of treadmills to 6 km or less, and limiting the use of the shower room) [63]. Consequently, many complaints were made by the stakeholders and clients of fitness centers. Moreover, controversy surrounded unscientific social distancing measures and COVID quarantine pass during the gradual recovery to normalcy [64]. In this manner, infectious diseases such as COVID-19 threaten the development and sustainability of fitness centers where activities are mostly performed in a face-to-face environment. Therefore, the government needs to expand support measures and implement related policies that consider the stable operation of fitness centers and the sustained economic activities of the workers in these facilities and guarantees the rights of the clients to exercise, even under the influence of external factors like infectious diseases.

After COVID-19, the topics related to fitness centers that appeared were Health and Beauty and Digital Smart Healthcare Services; these topics have a common theme with the topic of health-related industries, which became an issue before COVID-19. The main keywords for Health and Beauty include bio, fitness, beauty, global, release, and healthcare, while the main keywords for Digital Smart Healthcare Service include healthcare, digital, medical, development, data, smart, and platform. These keywords indicated an increased interest in health and beauty, which are the essential values provided by the fitness industry even before the COVID-19 pandemic, as well as an increasing prevalence of digital smart healthcare services, which are emerging as new technologies. In reality, fitness centers cannot avoid the impact of the prolonged COVID-19 pandemic in the short term. However, in the long term, consumers will spend more money on exercise and staying physically fit (e.g., diet), and the related market will experience a boom. This phenomenon is called the Dumbbell economy [65], and it has been reported that the related industry will continue to grow as the Dumbbell economy expands. In addition, according to the Worldwide Survey of Fitness Trends for 2021 [21], wearable technologies (no. 2) are a trend in the fitness industry owing to the development of related digital industries. Furthermore, new trends emerged due to major changes in the health and fitness industry after the outbreak of COVID-19, including online training (no. 1), outdoor activities (no. 4), virtual training (no. 6), and mobile exercise apps (no. 12). However, group training fell drastically in 2021 due to the closure of gyms and recommendations to limit group gatherings. Therefore, fitness trends worldwide are rapidly changing due to the COVID-19 pandemic to ensure people may be physically active by providing various services using online-based and digital equipment, as well as conventional face-to-face group fitness programs. However,

it was not possible to quickly determine the changes in fitness trends using news media reports and articles. Considering this limitation, future research should utilize social media data, which can be used to more directly access consumer information and quickly analyze market trends [66,67] to examine changes in fitness trends and the public's response to these changes. According to our research, no study had been conducted to examine the trends of major topics in fitness centers before and after COVID-19 through topic modeling. Therefore, it is difficult to discuss the results of this study considering the results of other studies. However, studies have been conducted in relation to COVID-19 in various research areas using topic modeling to derive related topics. These studies include higher education [68], health communication [10], and psychosocial stressors [11]. Therefore, the results of this study can be used to verify the usefulness of topic modeling in the analysis of news articles related to fitness-center-related articles.

Third, the sentiment analysis results of news articles related to fitness centers showed that the frequency of positive words decreased while the frequency of negative words increased by approximately three times after the outbreak of COVID-19. In particular, negative words were more prevalent than positive words after COVID-19. Negative words appeared in the descending order of violence, danger, charge, argument, damage, problem, and downturn before COVID-19. However, after COVID-19, negative words appeared in the order of infection, virus, disease, infectious disease, ban, declaration, limit, and discontinuation. This result indicated that before the outbreak of COVID-19, the press and public complained about the overall difficulties they encountered with fitness centers, including assault by fitness center personnel, membership fee refund issues, and problems with transferring memberships. Furthermore, an article studied by Cho [6] about problems in the fitness industry reported that while the rapid growth of the domestic fitness industry has increased the range of services available to participants, problems have continuously emerged, such as excessive competition between businesses, trainers' lack of expertise, and decline in the quality of services provided due to lack of expertise.

The health and fitness industry are considered a major industry that affects people's quality of life in terms of health, medical care, and welfare [69]. Following the COVID-19 pandemic, fitness centers had to operate their businesses in a limited manner or suspend their operations due to the preventive measures against COVID-19 enforced by the government, directly leading to restricting people from participation in sports activities. As a result, there was public concern about these preventive measures in terms of their negative effects on society in general, as well as public health which is the essential value provided by sports activities. Moreover, it has been reported that infectious diseases like COVID-19 have had a wide range of psychological and social impacts on individuals and local communities, and individuals experience the fear of being infected by such viruses [70]. According to the results of a public awareness survey on COVID-19, people experience anxiety (60.2%), fear (16.7%), shock (10.9%), and anger (6.7%) due to COVID-19-related news [71]. In particular, press reports about negative factors related to indoor sports facilities include group infection, the indoor spread of the virus, and viral infection if the air is not ventilated, even if a 2 m distance is maintained [72], as well as the possibility of transmission by sweaty hands [27]. Considering these findings, excessive reporting on viral infections by media platforms only increases public concern and anxiety [73]. Therefore, information should not be reported indiscriminately such that people may become excessively anxious. Instead, it is important to encourage people to selectively acquire correct knowledge and information, such as guidelines on how to use indoor sports facilities safely, how to wear a mask properly, and proper hand washing techniques. In addition, people are restricted from participating in exercise and physical activities due to the excessive restrictions imposed on indoor sports facilities [74]. Therefore, to encourage the public to participate in physical activities safely, it is necessary to devise future-oriented policies and measures to establish standards for preventive measures, operations, and stable statuses for the safe operation of indoor sports facilities. This is the first study to use big data analysis to explore the variational trends in major issues and press reports

(positive and negative sentiments) for fitness centers in South Korea before and after the outbreak of COVID-19. However, there are still some limitations in this study. First, the news articles we used contained data for South Korea concerning domestic fitness trends. Therefore, in future research, it is necessary to collect and analyze media reports for the fitness industry worldwide to examine overall trends. Second, this study analyzed social issues and trends related to fitness centers using only news articles. However, social media data, such as Twitter, Instagram, and Facebook, should be collected and analyzed in future studies because these data are effective in tracking consumer emotions [67]. Third, this study divided the collected data into two two-year periods—before and after the outbreak of COVID-19—and analyzed the collected data to examine the variational trends in the main topics related to fitness centers before and after COVID-19. Further studies need to be conducted using dynamic topic modeling reflecting the progress of time by identifying the topics of concern in a time series [75]. These studies may be used to determine the variation of topics over time (e.g., before, during, and after the COVID-19 pandemic). Lastly, we used topic modeling based on LDA to cluster meaningful topics from a large amount of unstructured news text data. Additional studies need to be conducted utilizing other algorithms and techniques (e.g., correlated topic modeling) to examine the correlation between topics and stochastic meaning, considering the correlations between topics. This study aimed to identify key issues and social discussions related to the fitness industry due to COVID-19 and derive implications for future improvements. Simultaneously, it was possible to identify issues that were relatively sparsely mentioned (e.g., gym owners, trainers, and coaches) and suggest focus factors for future studies. Therefore, it is expected that the results derived from this study may be used to devise a plan to further develop the fitness industry and provide foundational data for related research and policies.

**Author Contributions:** M.R. and H.P. contributed to the literature review and supervision. M.R. wrote the original draft and prepared and edited the original article. Y.C. developed the database, and H.P. analyzed and interpreted the data. All authors have read and agreed to the published version of the manuscript.

**Funding:** This research received no external funding.

**Institutional Review Board Statement:** Not applicable.

**Informed Consent Statement:** Not applicable.

**Data Availability Statement:** The data presented in this study are available upon request from the corresponding author.

**Conflicts of Interest:** The authors declare no conflict of interest.

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
