# Peer review of "Analysis of Issues in Fitness Centers through News Articles before and after the COVID-19 Pandemic in South Korea: Applying Big Data Analysis"

_sustainability, doi:10.3390/su15032660_

Round 1

Reviewer 1 Report

Manuscript title: Analysis of Issues in Fitness Centers Through News Articles before and after the COVID-19 Pandemic in South Korea: Applying Big Data Analysis. Overall, the subject matter of this paper is modern, interesting, and useful to the general reader. The authors wrote it concisely, to the point, and well supported by literature. However, it must be admitted that the content of the Research Findings is too short. The authors should consider expanding it so that the reader can see enough detail and understand it a bit more. I am very satisfied with this report. as it will be used as an important database for reporting and planning a policy in the field of fitness It is necessary to use a large database. to make a decision.

Reviewer 2 Report

Overall, this is a good research. The technological advance to me is quite nice and aimed to extract the main keywords 81 and topics from news articles about fitness centers related to COVID-19 and analyze related issues using big data analysis. But on the other hand, the paper requires minor adjustments on the following comments:

1. What is the contribution of the study must be added at the end introduction section?

2. The impact of using another dataset.

4. There are other Algorithms and Techniques used in Improving Topic Modeling are Latent Semantic Analysis, Correlated Topic Modeling, and Probabilistic Latent Semantic Analysis other than Latent Dirichlet Allocation, (LDA). While TF and ITF are very basic for data analysis

5. Following new references should be added in the paper where similar methodologies and dicu are applied and by studying and adding these will improve the quality of this paper:

  1. Alfaifi, A. A., & Khan, S. G. (2022). Utilizing Data from Twitter to Explore the UX of “Madrasati” as a Saudi e-Learning Platform Compelled by the Pandemic. Arab Gulf Journal of Scientific Research, 39(3), 200-208. doi:10.51758/AGJSR-03-2021-0025
  2. Khan, S. (2021). Visual Data Analysis and Simulation Prediction for COVID-19 in Saudi Arabia Using SEIR Prediction Model. International Journal of Online Biomedical Engineering, 17(8). doi:10.3991/ijoe.v17i08.20099
  3. A. U. Haq, J. P. Li, S. Ahmad, S. Khan, M. A. Alshara, and R. M. Alotaibi, "Diagnostic approach for accurate diagnosis of COVID-19 employing deep learning and transfer learning techniques through chest X-ray images clinical data in E-healthcare," Sensors, vol. 21, no. 24, p. 8219, 2021.
  4. Tayyab, M., Hussain, A., Alshara, M. A., Khan, S., Alotaibi, R. M., & Baig, A. R. (2022). Recognition of Visual Arabic Scripting News Ticker from Broadcast Stream. IEEE Access, 10, 59189 - 59204. doi:10.1109/ACCESS.2022.3179366

Reviewer 3 Report

I want to thank the authors for the opportunity to read their work. The article “Analysis of Issues in Fitness Centers Through News Articles before and after the COVID-19 Pandemic in South Korea: Applying Big Data Analysis. represents a valuable contribution to various disciplines, from political sciences, sociology, and international relations to communication studies, among other related disciplines. Therefore, the following comments are driven to increase the overall quality, clarify some points and offer a constructive review.

1. From a more pragmatic view, the authors should increase the justification for why they decided to analyze news pieces. The justification (in the Introduction) on Covid-19, fitness centers, and health is well-fitted. However, the decision on news effects on society appears only in a short sentence in the literature review. It’s not necessary to re-do all the Introduction section. My recommendation goes to incorporate any media/communication reference on it. Media effects on society is a strong and attractive field for both media scholars and sociologists and political science scholars. Thus, this can make your article more appealing to a broad audience. Complementing it could also be adjusted in the abstract. In short, it is to highlight with some reference what you do indeed alongside your work.    

2. In the Literature review: section 2.3 “Big data analysis in the health industry,” the title does not reflect what you encompass in it. Hence, I suggest two different possibilities: a) create a subsection on big data and news content analysis, developing in a few lines what you say briefly on lines 202, 203, and 216, and, thereafter, link it with a 2.4 section maintaining only the references on big data analysis in the health industry; b) maintain the section 2.3 changing the title to include “news” on it and re-organize the paragraphs below. Both must be as simple as you can. It’s not to re-do all this literature review section. It is more to make it more readable. For sure, this helps the discussion level to be stronger as well.  

3. Limitations of the study: we suggest including some methodological limitations (line 580 and ahead), which means incorporating the scope and limits of big data analysis of news itself (not only missing’s and future development, which are okay!).       

These minor reviews can make the article more soundness to a broader audience as well as can collaborate with it in general.  

Round 2

Reviewer 2 Report

The manuscript has been sufficiently improved to warrant publication in Sustainability